# PROTAC-Mediated Dual Degradation of BCL-xL and BCL-2 Is a Highly Effective Therapeutic Strategy in Small-Cell Lung Cancer

**DOI:** 10.3390/cells13060528

**Published:** 2024-03-17

**Authors:** Sajid Khan, Lin Cao, Janet Wiegand, Peiyi Zhang, Maria Zajac-Kaye, Frederic J. Kaye, Guangrong Zheng, Daohong Zhou

**Affiliations:** 1Department of Biochemistry & Structural Biology, Long School of Medicine, University of Texas Health Science Center at San Antonio, San Antonio, TX 78229, USA; 2Mays Cancer Center, University of Texas Health Science Center at San Antonio, San Antonio, TX 78229, USA; 3Department of Pharmacodynamics, College of Pharmacy, University of Florida, Gainesville, FL 32610, USA; 4Department of Medicinal Chemistry, College of Pharmacy, University of Florida, Gainesville, FL 32610, USA; 5Department of Anatomy & Cell Biology, College of Medicine, University of Florida, Gainesville, FL 32610, USA; 6Division of Hematology and Oncology, Department of Medicine, College of Medicine, University of Florida, Gainesville, FL 32610, USA

**Keywords:** BCL-xL, BCL-2, PROTAC, apoptosis, small-cell lung cancer

## Abstract

BCL-xL and BCL-2 are validated therapeutic targets in small-cell lung cancer (SCLC). Targeting these proteins with navitoclax (formerly ABT263, a dual BCL-xL/2 inhibitor) induces dose-limiting thrombocytopenia through on-target BCL-xL inhibition in platelets. Therefore, platelet toxicity poses a barrier in advancing the clinical translation of navitoclax. We have developed a strategy to selectively target BCL-xL in tumors, while sparing platelets, by utilizing proteolysis-targeting chimeras (PROTACs) that hijack the cellular ubiquitin proteasome system for target ubiquitination and subsequent degradation. In our previous study, the first-in-class BCL-xL PROTAC, called DT2216, was shown to have synergistic antitumor activities when combined with venetoclax (formerly ABT199, BCL-2-selective inhibitor) in a BCL-xL/2 co-dependent SCLC cell line, NCI-H146 (hereafter referred to as H146), in vitro and in a xenograft model. Guided by these findings, we evaluated our newly developed BCL-xL/2 dual degrader, called 753b, in three BCL-xL/2 co-dependent SCLC cell lines and the H146 xenograft models. 753b was found to degrade both BCL-xL and BCL-2 in these cell lines. Importantly, it was considerably more potent than DT2216, navitoclax, or DT2216 + venetoclax in reducing the viability of BCL-xL/2 co-dependent SCLC cell lines in cell culture. In vivo, 5 mg/kg weekly dosing of 753b was found to lead to significant tumor growth delay, similar to the DT2216 + venetoclax combination in H146 xenografts, by degrading both BCL-xL and BCL-2. Additionally, 753b administration at 5 mg/kg every four days induced tumor regressions. At this dosage, 753b was well tolerated in mice, without observable induction of severe thrombocytopenia as seen with navitoclax, and no evidence of significant changes in mouse body weights. These results suggest that the BCL-xL/2 dual degrader could be an effective and safe therapeutic for a subset of SCLC patients, warranting clinical trials in future.

## 1. Introduction

Small-cell lung cancer (SCLC) is a highly aggressive malignancy with an overall 5-year survival rate of below 7% [1]. It remains a formidable clinical challenge given the lack of effective targeted therapies and acquired resistance to conventional platinum-based chemotherapy, i.e., the doublet of etoposide with cisplatin or carboplatin [1,2,3,4]. Despite advances in our understanding of the molecular landscape of SCLC, therapeutic breakthroughs have been limited. Recently, a PD-L1 inhibitor (atezolizumab) in combination with carboplatin and etoposide has been approved to treat extensive stage (advanced) SCLC [5]. However, this chemoimmunotherapy regimen has been shown to promote only a modest improvement in overall survival of patients compared to the chemotherapy alone [3,6]. In June 2020, lurbinectedin, which works by inhibiting RNA polymerase II, was approved by the FDA for the treatment of metastatic SCLC patients who show continuing disease progression upon or after platinum-based chemotherapy [7,8]. However, lurbinectedin showed suboptimal responses in relapsed SCLC [9]. Thus, there is a critical unmet need for newer therapeutic strategies to effectively treat SCLC.

The BCL-2 family of proteins, including both pro-apoptotic and anti-apoptotic (or pro-survival) classes, are crucial regulators of the intrinsic (mitochondrial) pathway of apoptosis [10,11]. They have emerged as pivotal biomarkers in the survival of SCLC cells [12,13,14,15]. Among the anti-apoptotic members of the BCL-2 family, BCL-2, BCL-xL, and MCL-1 have garnered significant attention due to their prominent roles in promoting cell survival, chemoresistance, and disease progression in SCLC [16,17,18,19,20,21]. While individual targeting of these proteins has shown promise in preclinical studies, the development of resistance mechanisms and compensatory signaling pathways has hampered the success of monotherapies [17,20,22,23,24]. In recent years, a growing body of evidence has suggested that simultaneous inhibition of BCL-xL and BCL-2 or MCL-1 may represent a more effective strategy to combat SCLC [12,16,18,21,25]. Indeed, we have recently reported that the combined targeting of BCL-xL and MCL-1 with a platelet-sparing BCL-xL proteolysis targeting chimera (PROTAC) degrader (DT2216) and an mTOR inhibitor (AZD8055) is effective in inhibiting tumor growth in SCLC preclinical models without causing the on-target toxicities associated with BCL-xL and MCL-1 inhibitors [26]. In the same study, we found that a large subset of SCLC cell lines is co-dependent on BCL-xL and BCL-2 for their survival, as evident from their high sensitivity to the BCL-xL/2 inhibitor navitoclax (formerly ABT263). Unfortunately, navitoclax causes dose-limiting thrombocytopenia through on-target inhibition of BCL-xL in platelets. The severe platelet toxicity presents a great challenge in the clinical translation of navitoclax and other BCL-xL inhibitors [27,28,29]. Therefore, developing strategies to reduce BCL-xL inhibition-induced platelet toxicity has been an important research area for more than a decade now [30]. One of the earliest approaches to reduce BCL-xL inhibition-induced platelet toxicity was to convert a potent BCL-xL inhibitor into a pro-drug such as APG-1252 [31]. The idea behind a pro-drug of APG-1252 is to minimize its impact on platelets in the bloodstream until it reaches the tumor site, where it is then converted into its active form. We have used the PROTAC technology to reduce navitoclax thrombocytopenia by converting it into a VHL E3 ligase targeted BCL-xL PROTAC, such as DT2216, because platelets are devoid of the VHL E3 ligase required for BCL-xL degradation [32]. In contrast, these PROTACs can efficiently degrade BCL-xL in tumor cells, which possess significantly high levels of VHL compared to platelets [32,33,34,35].

Through extensive structural optimizations, we have developed new PROTACs that are capable of degrading both BCL-xL and BCL-2 [36]. The lead first-in-class BCL-xL and BCL-2 dual degrader, named 753b, was shown to more potently degrade BCL-xL compared to DT2216, with concomitant degradation of BCL-2. This was due to the accessibility of a key lysine residue in BCL-2 that was not accessible by DT2216, as well the formation of a more stable ternary complex between VHL E3 ligase, 753b, and BCL-xL or BCL-2. Due to degrading both the proteins, 753b was found to be more potent in killing of BCL-xL/2 co-dependent cancer cells [36].

In this study, we capitalized on the synergistic effects of degrading BCL-xL and BCL-2 in SCLC using a single degrader, i.e., 753b [36]. Since a large subset of SCLCs are characterized by high BCL-xL and BCL-2 mRNA and protein expression, we evaluated 753b in comparison to DT2216, navitoclax, and DT2216 + venetoclax combination in terms of efficiency and specificity for BCL-xL and BCL-2 degradation, as well as the viability of SCLC cells. Next, we evaluated the anti-tumor efficacy of 753b in NCI-H146 (hereafter referred to as H146) SCLC xenograft models at two different dosing frequencies and tumor sizes. The degradation of BCL-xL and BCL-2 following treatment with 753b was confirmed in tumor tissues. Collectively, our results show that 753b is significantly more potent than DT2216, navitoclax, and the DT2216 + venetoclax combination in killing BCL-xL/2 co-dependent SCLC cells via degradation of both the proteins. In vivo, 753b requires a significantly lower dosage to elicit similar effects to DT2216 + venetoclax, and is capable of regressing larger H146 xenograft tumors. 

## 2. Materials and Methods

### 2.1. Cell Lines and Culture 

The SCLC cell lines, except H146 (hereafter, all of the SCLC cell lines in this article are referred to without the prefix ‘NCI’), were obtained from the NCI-Navy Medical Oncology source supply as stated in our previous publication [26]. H146 cells were obtained from the American Type Culture Collections (ATCC, Manassas, VA, USA). The cell lines were cultured as reported previously [26]. The stocks of the cell lines were STR profiled by NIH or the ATCC. WI38 normal lung fibroblasts were obtained from ATCC and were cultured in a high-glucose DMEM medium (Cat. No. 12430062, Thermo Fisher, Waltham, MA, USA), supplemented with fetal bovine serum (FBS) and penicillin-streptomycin. All cultures were tested for mycoplasma negativity using the MycoAlert Mycoplasma Detection Kit (Cat. No. LT07–318, Lonza, Basel, Switzerland). The cell lines were maintained in a humidified incubator at 37 °C and 5% CO_2_.

### 2.2. Chemical Compounds

DT2216 and 753b were provided by Dr. Guangrong Zheng’s laboratory (University of Florida, Gainesville, FL, USA), after their synthesis according to the previously described protocol [32,36]. Navitoclax (Cat. No. S1001) and venetoclax (Cat. No. S8048) were purchased from SelleckChem (Houston, TX, USA). The compounds were dissolved at 10 mM stock concentrations in DMSO for in vitro assays. The details of in vivo formulations are provided in the ‘Tumor xenograft studies’ method.

### 2.3. Cell Viability Assays

Cells were plated in 96-well plates at a density of 5 × 10^3^ (adherent cells) or 5 × 10^4^ (suspension cells) per well, and then treated the following day (adherent cells) or the same day (suspension cells) in nine-point three-fold serial dilutions of the drugs in three to six replicates. The cell viability was measured by adding MTS reagent (Cat. No. G-111, Promega, Madison, WI, USA) according to the manufacturer’s protocol and as described previously [26,32]. The absorbance was recorded at 490 nm using Biotek’s Synergy Neo2 multimode plate reader (Biotek, Winooski, VT, USA). The half-maximal inhibitory concentration (IC_50_) values were determined using GraphPad Prism 10.2.1 software (GraphPad Software, La Jolla, CA, USA).

### 2.4. Immunoblotting

Cell/tissue lysate samples were prepared and immunoblotting analysis was performed according to our previously reported protocol [26,32]. After protein transfer, the PVDF membranes were probed with primary antibodies overnight at 4 °C, and then incubated with horseradish peroxidase (HRP)-linked secondary antibody for 1–2 h at room temperature. Finally, the enhanced chemiluminescence (ECL) substrate (Cat. No. WBKLS0500, Millipore Sigma, Billerica, MA, USA) was used for signal detection in a ChemiDoc MP Imaging System (Bio-Rad, Hercules, CA, USA). The densitometry of blots was quantified using ImageJ 1.53k software. The details of the primary antibodies are given in the Appendix A. 

### 2.5. Tumor Xenograft Studies

Five week old female CB-17 SCID-beige mice were purchased from the Charles River Laboratories (Wilmington, MA, USA). A total of 5 × 10^6^ H146 cells mixed with 100 μL of Matrigel-RPMI mixture (1:1) were injected subcutaneously (s.c.) into the right flank region of each mouse, as described previously [26,32]. When palpable tumors appeared, they were measured twice a week with digital calipers and the recorded data were converted to tumor volume (mm^3^) format using the formula (Length × Width^2^ × 0.5). When the tumors reached ~150 mm^3^ (for the tumor inhibition study) or ~500 mm^3^ (for the tumor regression study), the mice were randomized based on tumor volume and started undergoing treatment with vehicle, 753b (5 mg/kg, once a week or every four days, i.p.), DT2216 (15 mg/kg, once a week, i.p.), or DT2216 plus venetoclax (50 mg/kg, 5 days a week, p.o.). The dosages of 753b, DT2216, and venetoclax were selected based on our previous studies [26,32,37]. DT2216 and 753b were formulated in 50% phosal 50 PG, 45% miglyol 810 N, and 5% polysorbate 80. Post-euthanasia, the tumors were harvested, lysed, and used for immunoblotting analysis. All the animal experiments were performed in accordance with the approved Institutional Animal Care and Use Committee (IACUC) protocol. 

### 2.6. Platelet Counts 

Mouse blood was collected in EDTA-treated tubes via the submandibular plexus. The blood was immediately used for platelet counts using a HEMAVET 950 FS hematology analyzer (Drew Scientific Inc., Miami Lakes, FL, USA). The data were expressed as number of platelets per µL of blood.

### 2.7. Statistical Analysis

The analysis of variance (ANOVA) test was performed for analysis of the means of three or more experimental groups. When ANOVA justified post-hoc comparisons between group means, the comparisons were made using Tukey’s multiple-comparisons test. A two-sided Student’s *t*-test was used for comparisons between the means of two groups. *p* < 0.05 was the cut-off for statistical significance. 

## 3. Results

### 3.1. 753b Is a Dual BCL-xL/2 Degrader in SCLC Cells

In our previous study, we conducted a BH3 mimetic screening, where SCLC cell lines were treated with selective inhibitors of BCL-2, BCL-xL and MCL-1 as well as a BCL-xL/2 dual inhibitor to determine their survival dependencies. Through this screening, we determined that SCLC cell lines are heterogenous in their dependence on BCL-2 family proteins, and we defined a subset of SCLC tumors that depend on both BCL-2 and BCL-xL for survival [26]. 

The BCL-xL/2 dual inhibitor navitoclax is highly effective in treating SCLC as evidenced by CancerRxGene data, where about 70% of the cell lines were found to be sensitive [38]. Similar results were obtained via BH3 mimetic screening performed by us in a previous study [26]. In that screening, we also found that about 45% of the tested SCLC cell lines are sensitive to BCL-xL-selective inhibitor A1155463, and about 70% are sensitive to navitoclax. These observations suggest that dual targeting of both BCL-xL and BCL-2 is a preferred therapeutic strategy for treating SCLC. Since navitoclax causes dose-limiting and on-target thrombocytopenia through inhibition of BCL-xL in platelets, it has not been further developed for clinical translation. To circumvent these challenges, we have recently identified a BCL-xL/2 dual degrader, named 753b, through extensive structural optimization of DT2216 (Figure 1a) [36]. However, the dual BCL-xL/2 degradation by 753b was not evaluated in SCLC cells in our previous proof-of-concept study. The 753b compound required shorter exposure times compared to DT2216 to induce BCL-xL degradation in SCLC cells, which typically grow in multicellular clusters (Figure 1b–d; Appendix A). Moreover, 753b exhibited higher cell line average D_max_ (% maximum degradation) and reduced average DC_50_ (concentration at which 50% degradation occurs) for BCL-xL in H146 and H211 cells compared to DT2216 (Figure 1e–g; Appendix A). Additionally, 753b induced substantial degradation of BCL-2 (D_max_: 26.3 to 62.7%) in SCLC cells (Figure 1b–g). Among the SCLC cell lines tested, H211 cells showed maximum BCL-2 degradation, followed by H1059 and H146, respectively. Notably, we observed moderate MCL-1 suppression at higher doses of 753b in H146 cells, likely due to MCL-1 cleavage by activated caspase-3, as reported previously [39,40,41]. Treatment with 753b resulted in no considerable degradation of BCL-w. These results suggest that 753b is an effective dual degrader of BCL-xL and BCL-2 in SCLC cells.

### 3.2. 753b Is Highly Potent to Kill BCL-xL/2-Dependent SCLC Cells

The dual BCL-xL/2 degradation by 753b induced robust apoptosis as indicated by increased caspase-3 and PARP cleavage in SCLC cells (Figure 1b–d). Specifically, 753b exerted 12-, 15- and five-fold higher potency than DT2216 to kill BCL-xL/2-dependent H146, H211, and H1059 cells, respectively. Moreover, it was two- and four-fold more potent than navitoclax against H146 and H211 cells, respectively (Figure 2a–c,f). We also tested the effect of 753b on the viability of BCL-xL/MCL-1-dependent H378 cells and WI38 normal lung fibroblasts, which were found to be mildly sensitive and resistant to 753b, respectively (Figure 2d–f). This suggests that 753b exerts BCL-xL/2-specific activity and possesses a wide therapeutic window. 

H146, H211, and H1059 cells were more sensitive to a combination of DT2216 with venetoclax compared to mono-targeting, confirming their co-dependence on BCL-xL and BCL-2 (Figure 3a–c). Therefore, we tested whether venetoclax could similarly enhance the efficacy of 753b. We found that venetoclax could only slightly enhance the efficacy of 753b (Figure 3d–f). This was expected because 753b cannot completely degrade BCL-2, so the slightly enhanced efficacy might have resulted from the inhibition of remaining BCL-2. However, 753b alone was equally effective as the combination of DT2216 and venetoclax against these SCLC cell lines. These results confirm that 753b is a dual BCL-xL/2 degrader that can be more potent against BCL-xL/2 co-dependent SCLC cells than DT2216 alone and does not require combination with venetoclax to effectively kill these cells. 

### 3.3. 753b Showed Similar Anti-Tumor Efficacy as DT2216 + Venetoclax by Dual BCL-xL/2 Degradation in H146 Xenograft Model

Finally, we tested the in vivo efficacy and safety of 753b using H146 xenograft models. In the first study, tumor bearing mice were treated with 5 mg/kg weekly dosing of 753b, compared with DT2216 at 15 mg/kg weekly alone or in combination with venetoclax (50 mg/kg, 5 days a week). Despite a three-fold lower dosing, 753b led to similar tumor growth inhibition as DT2216 initially for five weeks, and from thereon, tumors grew slower in 753b-treated mice compared to DT2216-treated mice. Tumor growth after five weeks of treatment was similar in 753b- and DT2216 + venetoclax-treated mice (Figure 4a). In addition, mice treated with 753b showed significantly enhanced survival compared to vehicle control (median survival: 104 vs. 32 days) and showed a trend towards extended survival compared to DT2216 (median survival: 104 vs. 94 days) (Figure 4b). The median survival in both the 753b- and DT2216 + venetoclax-treated groups was 104 days. None of these treatments caused any significant changes in mouse body weights (Figure 4c). We also measured platelet levels one day after the first dose and three days after the third weekly dose. We observed that the platelet reduction one day after the first dose was more pronounced in 753b-treated mice than DT2216-treated mice, but similar reductions were observed three days after the third dose. Addition of venetoclax to DT2216 caused no further reduction in platelets. Notably, these reductions in platelets did not reach clinically dangerous levels, i.e., below 1 × 10^5^ per μL of blood, and thus were well tolerated in mice (Appendix A). These results suggest that 753b is a more potent antitumor agent than DT2216 and shows similar efficacy as the combination of DT2216 + venetoclax against BCL-xL/2 co-dependent SCLC.

We wondered whether 753b could induce degradation of both BCL-xL and BCL-2 in tumors, as observed in vitro. Indeed, 753b showed significant degradation of both BCL-xL and BCL-2 in excised xenograft tumors. As expected, DT2216 selectively degraded BCL-xL, without significant effect on BCL-2. Neither 753b, nor DT2216 showed any significant effect on MCL-1. These data suggest that the enhanced anti-tumor efficacy of 753b compared to DT2216 is derived from dual degradation of BCL-xL and BCL-2 in tumors (Figure 4d,e). 

### 3.4. 753b Treatment Induced Tumor Regressions in H146 Xenograft Model

We also investigated whether 753b can regress larger established tumors in mice. For this, mice were treated every four days with a 5 mg/kg dose of 753b when the average tumor sizes were more than 500 mm^3^. Interestingly, 753b caused significant tumor regressions in these mice. These regressions sustained for about three weeks, and then tumors started to slowly regrow (Figure 5a). The excised tumors were significantly smaller in 753b-treated mice compared to vehicle-treated mice (Figure 5b,c). At this dosage, 753b caused no significant changes in mouse body weights, suggesting this dosing regimen is tolerated in mice (Figure 5d).

## 4. Discussion

The BCL-2 family of anti-apoptotic proteins, such as BCL-xL, BCL-2, and MCL-1, represent promising therapeutic targets for SCLC, a malignancy characterized by its aggressive nature and resistance to standard treatment modalities [1,2,3,4,12,13,14,15]. Therefore, the targeting of BCL-xL and BCL-2 with small molecule inhibitors has been explored as a therapeutic strategy in SCLC, since the discovery of navitoclax [1,16,17,18,19,20,21]. However, the only FDA approved BCL-2 family protein inhibitor for the treatment of relapsed/refractory chronic lymphoblastic leukemia (CLL) and acute myelogenous leukemia (AML) is an alternate BCL-2-selective agent, venetoclax [42,43]. The single agent efficacy of venetoclax is limited in SCLC because only a very small subset of SCLCs exhibit dependence on BCL-2 expression for survival, while a large subset of SCLC samples exhibit co-dependence on BCL-xL and BCL-2 or MCL-1 [12,13,16]. Therefore, a more promising strategy to co-target BCL-xL and MCL-1 with DT2216 and AZD8055, respectively, in SCLC has been recently reported by us [26]. In the current study, we focus on co-targeting BCL-xL and BCL-2. Since the clinical drug development of the BCL-xL/2 dual inhibitor navitoclax has been hindered by dose-limiting and on-target thrombocytopenia due to BCL-xL inhibition in platelets [27,28,29], we have recently converted navitoclax into platelet-sparing BCL-xL PROTAC degraders using VHL E3 ligase for their activity, which significantly reduced platelet toxicity [32]. This is because VHL is minimally expressed in platelets, limiting the effect on platelets and leading to selective killing of tumor cells [32,33,34,35].

Given the high potency of navitoclax, we proposed that a dual BCL-xL and BCL-2 degrader could have similar or improved efficacy with reduced platelet toxicity in SCLC compared to navitoclax. Our group has recently reported a BCL-xL/2 dual degrader, named 753b, that shows improved antileukemic activity compared to BCL-xL-selective degraders such as DT2216 [36]. Therefore, we utilized 753b to test our hypothesis in SCLC. 753b was found to be five- to 15-fold more potent than DT2216 in killing BCL-xL/2 co-dependent SCLC cells. Moreover, it was two- to four-fold more potent than navitoclax likely due to the catalytic mechanism of PROTACs. Furthermore, 753b induced deeper and rapid degradation of BCL-xL at significantly lower concentrations than DT2216, and concomitantly led to BCL-2 degradation. However, the extent of BCL-2 degradation by 753b was cell line-dependent and was seen at slightly higher concentrations compared to BCL-xL degradation. Notably, BCL-xL degradation at higher concentrations of 753b in H146 cells was suppressed. This phenomenon of suppressed protein degradation at higher concentrations of PROTACs is known as the “hook effect”, where binary complexes between PROTAC and the target protein or E3 ligase compete with PROTAC-target protein-E3 ligase ternary complex formation. Due to the reduced ternary complex formation, target protein degradation is suppressed at higher concentrations of some PROTACs in a cell-line dependent manner [44,45,46,47].

Furthermore, we elucidated the downstream signaling events triggered by the dual degradation of BCL-xL and BCL-2. Administration of 753b led to a strong activation of caspase-3 and PARP, demonstrating a pronounced shift towards apoptotic pathways in SCLC cells. The high cell death at supra-IC_50_ concentrations of 753b led to MCL-1 suppression, possibly through a caspase 3-mediated cleavage. This is because MCL-1 is one of the substrates of caspase-3 [39,40,41]. The observed increase in caspase activation and cleavage of key apoptotic substrates reinforces the notion that co-targeting BCL-xL and BCL-2 for degradation induces a robust pro-apoptotic response in SCLC cells, ultimately leading to reduced cell viability. 

Further evaluations in H146 xenograft models demonstrated that 753b requires threefold lower dosage (5 mg/kg once a week) to elicit similar antitumor activity to the combination of DT2216 + venetoclax. However, the tumors started slowly regrowing after five weeks of treatment, though the regrowth was slower in 753b- and DT2216 + venetoclax treated mice compared to DT2216-treated mice. Additionally, 753b at this dosage led to a trend of extending median survival compared to DT2216; however, the difference did not reach statistical significance. The 753b was tolerated in mice with no measurable toxicity, including no significant body weight loss nor induction of severe platelet toxicity. The platelet reductions after 753b treatment never reached below 1 × 10^5^ per µL of blood, which is considered to be safe [26,32]. Despite the lower dosing, 753b exerted significant BCL-xL and BCL-2 degradation in tumors collected at the end of treatment, which further supports the data showing comparable antitumor activity of 753b to the DT2216 + venetoclax combination. Treatment with 753b and DT2216 + venetoclax also resulted in moderate, but non-significant, reductions in MCL-1 levels. These results also suggest that the tumor regrowth in 753b- and DT2216-treated mice was not due to their inability to degrade BCL-xL and/or BCL-2 or compensatory elevation of MCL-1. Therefore, it would be crucial to elucidate the mechanisms responsible for tumor regrowth after a period of tumor statis upon 753b- and DT2216 treatment for designing combination strategies to treat SCLC more effectively in future studies.

Though it is a common laboratory practice to treat mice when their tumors are small in size (100–200 mm^3^), we are aware that the tumor sizes in human patients are larger at the initiation of therapy. Therefore, we tested whether 753b could inhibit or regress larger tumors in mice in a separate experiment using the H146 xenograft model. We initiated 753b treatment when the tumors were >500 mm^3^. Since the tumors were significantly larger, we treated mice with 5 mg/kg of 753b every four days. At this dosage, 753b was found to be efficient in regressing these larger H146 xenograft tumors. The tumor regressions sustained for three weeks and then the tumors started slowly growing, leading to significant tumor growth delay without causing any changes in mouse body weights. 

Our findings further support the hypothesis that dual targeting of BCL-xL and BCL-2 results in a synergistic pro-apoptotic effect, effectively disrupting the survival machinery of SCLC cells. This synergism is consistent with previous studies highlighting the compensatory mechanisms and functional redundancy within the BCL-2 family proteins, which may explain the limited success of monotherapies targeting either BCL-xL or BCL-2 alone [17,20,22,23,24]. Additionally, our study indicates that delayed resistance may arise in response to treatment with a dual BCL-xL/2 degrader, which may not be due to non-degradation of BCL-xL and BCL-2. This might be attributed to the capacity of cancer cells to adapt to therapeutic pressures through processes such as compensatory activation of other pro-survival pathways. Therefore, future research should focus on identifying strategies to overcome potential resistance mechanisms, such as exploring combination therapies.

In conclusion, our study provides compelling evidence for the therapeutic efficacy of co-targeting BCL-xL and BCL-2 using PROTAC degraders in SCLC. Early-phase clinical trials evaluating the safety and efficacy of navitoclax in SCLC patients have shown encouraging efficacy, underscoring the translational potential of this therapeutic approach. However, navitoclax showed severe reductions in platelet levels in patients, which hinders its clinical translation [27]. Therefore, the BCL-xL/2 dual degraders warrant clinical testing, due to their high potential for enhanced efficacy and/or reduced toxicity. The ASCL1-high molecular subtype of SCLC has been shown to be sensitive to BCL-xL/2 inhibition with navitoclax [48]. Therefore, this SCLC subtype could be sensitive to 753b treatment as well. Immunohistochemical profiling of four transcriptional regulators in SCLC, i.e., ASCL1, NEUROD1, POU2 F3, and YAP1 could be performed to stratify patients who can specifically benefit from 753b.

We have demonstrated a potent pro-apoptotic response and laid the foundation for further clinical development of this innovative therapeutic strategy. As we move forward, a multidisciplinary approach encompassing preclinical investigations, translational research, and clinical trials will be instrumental in realizing the full potential of dual BCL-xL/2 degradation in improving outcomes for SCLC patients.

## Figures and Tables

**Figure 1 cells-13-00528-f001:**
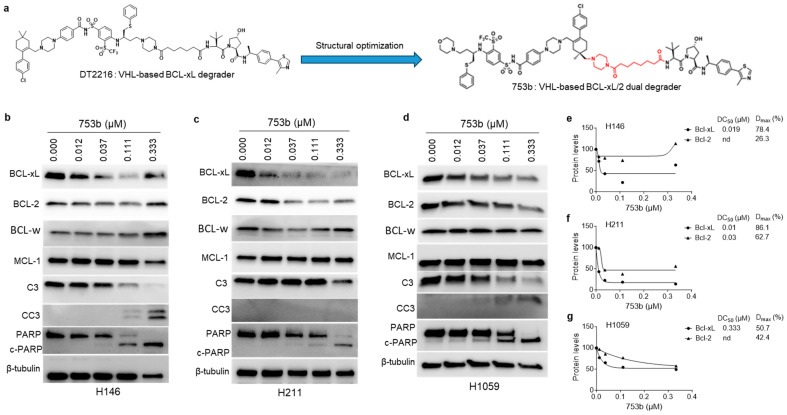
753b degrades both BCL-xL and BCL-2 leading to apoptosis in SCLC cells. (**a**) Chemical structures of DT2216 and 753b. Linker in 753b is highlighted in red. (**b**–**d**) Immunoblot analyses of BCL-X_L_, BCL-2, BCL-w, MCL-1, caspase 3 (C3), cleaved caspase 3 (CC3), PARP and cleaved (c)-PARP in SCLC H146 (**b**), H211 (**c**), and H1059 cells (**d**) after they were treated with indicated concentrations of 753b for 24 h. The β-tubulin was used as an equal loading control. (**e**–**g**) Densitometric analysis of BCL-xL and BCL-2 immunoblots in H146 (**e**), H211 (**f**), and H1059 cells (**g**) showing DC_50_ and D_max_ values for each protein. DC_50,_ concentration required to degrade 50% of the protein; D_max_, maximum degradation in percentage; nd, not determined.

**Figure 2 cells-13-00528-f002:**
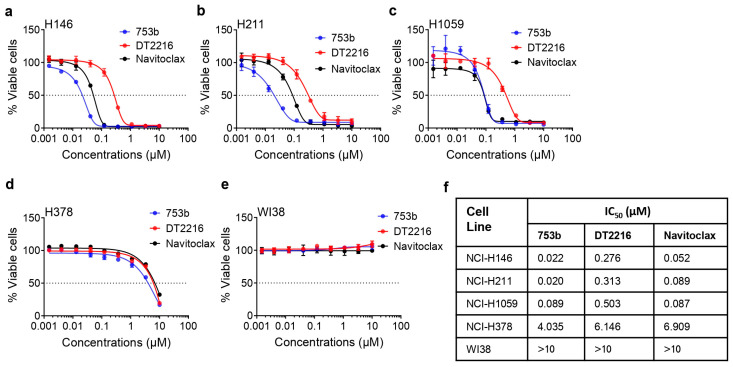
753b is more potent than DT2216 or navitoclax to kill BCL-X_L_/2-dependent SCLC cells. (**a**–**e**) Viability of H146 (**a**), H211 (**b**), H1059 (**c**), and H378 (**d**) SCLC cells, and WI38 normal lung fibroblasts (**e**), after they were treated with increasing concentrations of 753b, DT2216, or navitoclax for 72 h. (**f**) IC_50_ values for 753b, DT2216, and navitoclax in SCLC cell lines and WI38 cells are tabulated.

**Figure 3 cells-13-00528-f003:**
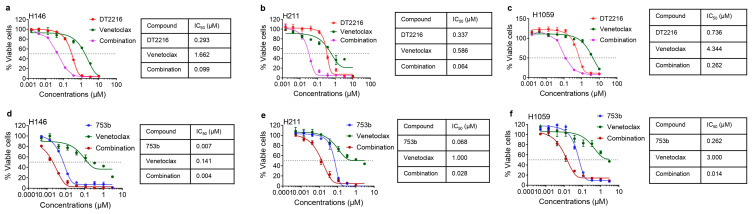
753b showed comparable or enhanced efficacy as compared to DT2216 + venetoclax combination to kill BCL-xL/2-dependent SCLC cells. (**a**–**c**) Viability of H146 (**a**), H211 (**b**), and H1059 cells (**c**) after they were treated with increasing concentrations of DT2216, venetoclax, or their 1:1 combination. IC_50_ values for the individual agents and combinations are tabulated. (**d**–**f**) Viability of H146 (**d**), H211 (**e**), and H1059 cells (**f**) after they were treated with increasing concentrations of 753b, venetoclax, or their 1:1 combination. IC_50_ values for the individual agents and combinations are tabulated.

**Figure 4 cells-13-00528-f004:**
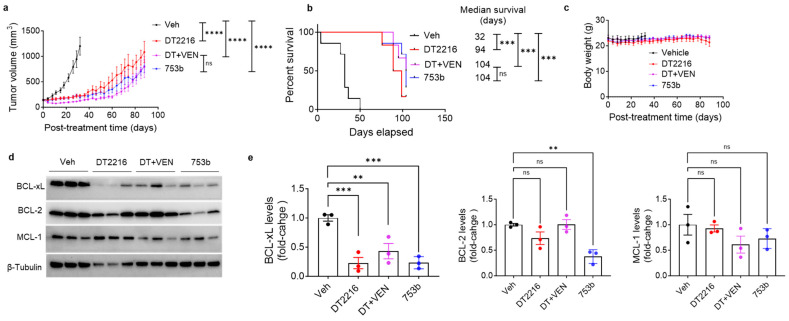
753b is more potent than DT2216 and similar to DT2216 + venetoclax for inhibiting growth of BCL-X_L_/2-dependent H146 xenograft tumors in mice. (**a**) Tumor volume changes in H146 xenografts after treatment with vehicle, DT2216 (15 mg/kg, weekly i.e., q7d, i.p.), a combination of DT2216 with venetoclax (50 mg/kg, 5 days a week, p.o.), or 753b (5 mg/kg, q7d, i.p.). Data are presented as mean ± SEM (*n* = 7, 6, 6, and 7 mice in vehicle, DT2216, DT2216 + venetoclax, and 753b groups, respectively, at the start of treatment). When the biggest tumor dimension reached 1.5 cm, the mice were sacrificed in accordance with IACUC protocol, and the remaining mice continued to be treated for up to 104 days. Tumor volume changes are shown up to post-treatment day 88, when 5 or more mice were alive in each treatment group. **** *p* < 0.0001, ns: not significant as determined by one-way ANOVA and Dunnett’s multiple comparisons test at post-treatment day 32. (**b**) Kaplan-Meier survival analysis of mice as treated in (**a**). Survival time was recorded at the tumor endpoint, i.e., biggest tumor dimension of 1.5 cm or more. The median survival time is shown on the right. *** *p* < 0.001, ns: not significant as determined by two-sided Student’s *t*-test. (**c**) Mouse body weight changes in H146 xenografts after treatment as in (**a**). Data are presented as mean ± SEM. (**d**) Immunoblot analysis of BCL-xL, BCL-2, and MCL-1 in H146 xenograft tumors two days after last treatment with vehicle, DT2216, DT2216 + venetoclax (DT + VEN), or 753b (*n* = 3 mice per group) as in (**a**). (**e**) Densitometric analysis of immunoblots in (**d**). ** *p* < 0.01, *** *p* < 0.001 compared to vehicle as determined by one-way ANOVA and Dunnett’s multiple comparisons test.

**Figure 5 cells-13-00528-f005:**
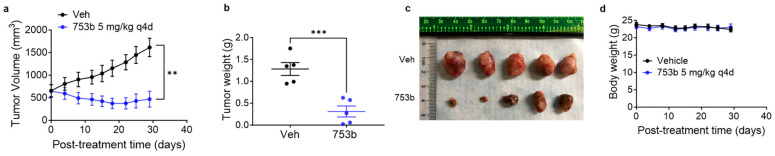
753b induces regressions of larger H146 xenograft tumors in mice. (**a**) Tumor volume changes in H146 xenografts after treatment with vehicle or 753b (5 mg/kg, every four days i.e., q4d, i.p.). Data are presented as mean ± SEM (*n* = 5 mice). ** *p* < 0.01 compared to vehicle as determined by two-sided Student’s *t*-test. (**b**), Tumor weights at the end of experiment in (**a**). Data are presented as mean ± SEM (*n* = 5 mice). *** *p* < 0.001 compared to vehicle as determined by two-sided Student’s *t*-test. (**c**) The images of excised tumors from (**a**). (**d**) Mouse body weight changes in H146 xenografts after treatment as in (**a**).

## Data Availability

All the relevant data are available in the main text or the Appendix A.

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
