# Peer review of "PROTAC-Mediated Dual Degradation of BCL-xL and BCL-2 Is a Highly Effective Therapeutic Strategy in Small-Cell Lung Cancer"

_cells, 2024, doi:10.3390/cells13060528_

Round 1
Reviewer 1 Report
Comments and Suggestions for Authors
In this study titled " PROTAC-mediated dual degradation of BCL-xL and BCL-2 is a 2 highly effective therapeutic strategy in small-cell lung cancer" by Sajid Khan et al., the authors investigated the efficacy of their novel dual degrader, 753b, targeting BCL-xL/2 in 3 BCL-xL/2 co-dependent SCLC cell lines and the H146 xenograft model. Their findings demonstrate that 753b effectively degrades both BCL-xL and BCL-2 proteins, leading to tumor regression in vivo without adverse effects such as severe thrombocytopenia and weight loss. These results suggest that the BCL-xL/2 dual degrader holds promise as a safe and effective therapeutic option for a specific subset of SCLC patients.
While the study presents an advancement in therapeutic strategy, it is essential to note the limitations of the research, including the narrow scope of cell lines and xenograft models tested. Expanding the in vitro panel to include a broader range of cell lines and incorporating patient-derived xenograft models in vivo would enhance the robustness of the findings and provide a more comprehensive understanding of the therapeutic potential of 753b.
Furthermore, it is crucial to delineate the characteristics of the subset of SCLC patients who would benefit most from BCL-xL/2 dual degradation therapy. This may involve identifying specific gene signatures or signaling pathways associated with BCL-xL/2 dependency within the SCLC population. Such delineation would facilitate personalized treatment approaches for this promising therapeutic strategy.
Author Response
Reviewer 1
In this study titled " PROTAC-mediated dual degradation of BCL-xL and BCL-2 is a highly effective therapeutic strategy in small-cell lung cancer" by Sajid Khan et al., the authors investigated the efficacy of their novel dual degrader, 753b, targeting BCL-xL/2 in BCL-xL/2 co-dependent SCLC cell lines and the H146 xenograft model. Their findings demonstrate that 753b effectively degrades both BCL-xL and BCL-2 proteins, leading to tumor regression in vivo without adverse effects such as severe thrombocytopenia and weight loss. These results suggest that the BCL-xL/2 dual degrader holds promise as a safe and effective therapeutic option for a specific subset of SCLC patients.
While the study presents an advancement in therapeutic strategy, it is essential to note the limitations of the research, including the narrow scope of cell lines and xenograft models tested. Expanding the in vitro panel to include a broader range of cell lines and incorporating patient-derived xenograft models in vivo would enhance the robustness of the findings and provide a more comprehensive understanding of the therapeutic potential of 753b.
Response: We thank the reviewer for his/her suggestion. We selected the three SCLC cell lines (H146, H211 and H1059) which are co-dependent on BCL-xL and BCL-2 for their survival based on our previous study (PMID 36588105). In that study, we performed a BH3 mimetic screening in 20 SCLC cell lines, where H146, H211 and H1059 were found to be dependent on both BCL-xL and BCL-2 for their survival, and therefore used for the current studies. We have confirmed the in vivo BCL-xL and BCL-2 degradation and efficacy of 753b in H146 xenograft tumor inhibition model (Fig. 4) and tumor regression model (Fig. 5) at weekly and twice a week dosing, respectively. We understand it would be important to evaluate the efficacy of 753b in patient-derived xenograft models of SCLC. However, these studies will require a longer time, and therefore, could not be performed at this time.
Furthermore, it is crucial to delineate the characteristics of the subset of SCLC patients who would benefit most from BCL-xL/2 dual degradation therapy. This may involve identifying specific gene signatures or signaling pathways associated with BCL-xL/2 dependency within the SCLC population. Such delineation would facilitate personalized treatment approaches for this promising therapeutic strategy.
Response: Recently, SCLC tumors have been classified into four molecular subtypes based on the relative expression of four transcriptional regulators, i.e., ASCL1, NEUROD1, POU2F3, and YAP1. ASCL1-high subtype has been shown to be sensitive to BCL-xL/2 inhibition with navitoclax (PMID 25267614). Therefore, ASCL1-high subset of SCLC could be sensitive to 753b treatment as well. Immunohistochemical profiling of these four biomarkers could be used for stratifying patients who can specifically benefit from 753b. We have briefly discussed this in the revised manuscript (Line 400-404)
Reviewer 2 Report
Comments and Suggestions for Authors
Review:
In this line of research, the authors wanted to demonstrate that PROTAC-mediated dual degradation of BCL-xL and BCL-2 is a highly effective therapeutic strategy in small-cell lung cancer. However, the reviewer is puzzled with the data shown in the current manuscript including the followings.
1. In Figure 1 b, the degradation of BCL-xL seems to be supressed at higher doses of 753b in H146 cells, what is the reason?
2. It was well known that BCL-xL and BCL-2 played prominent roles in promoting cell survival in various cell type. Besides detecting lung cancer cell viability, does 753b affect the occurrence and development of small cell lung cancer by affecting cell proliferation, apoptosis, and other abilities? Or use multiple experimental methods to prove your conclusion.
3. 753b showed comparable efficacy as DT2216+venetoclax combination to kill BCL-XL/2-dependent SCLC cells. So what are the advantages of 753b compared with DT2216+venetoclax?
4. In Figure 1 d, image of BCL-w IB was poor, a more high-quality image should be provided.
5. The last chart of of Fig4e was mislabeled?
6. In this paper, the drug 753b was verified by in vivo experiments. The authors should include the detail to determine the drug dosage in experimental method part.
Comments on the Quality of English LanguageMinor editing in English is required.
Author Response
Reviewer 2
In this line of research, the authors wanted to demonstrate that PROTAC-mediated dual degradation of BCL-xL and BCL-2 is a highly effective therapeutic strategy in small-cell lung cancer. However, the reviewer is puzzled with the data shown in the current manuscript including the followings.
- In Figure 1 b, the degradation of BCL-xL seems to be supressed at higher doses of 753b in H146 cells, what is the reason?
Response 1: We thank the reviewer for noticing it. This phenomenon of suppressed degradation at higher concentrations of PROTACs is known as “hook effect” when binary complexes between PROTAC and a target protein or E3 ligase compete with PROTAC-target protein-E3 ligase ternary complex formation at higher concentrations. Due to the reduced ternary complex formation, target protein degradation is suppressed at higher concentrations of some PROTACs. This is commonly observed with PROTACs in a cell-line dependent manner (PMID 29118097; 31846828; 28595007; 32292566). We have discussed this briefly in the revised manuscript (Line 338-344).
- It was well known that BCL-xL and BCL-2 played prominent roles in promoting cell survival in various cell type. Besides detecting lung cancer cell viability, does 753b affect the occurrence and development of small cell lung cancer by affecting cell proliferation, apoptosis, and other abilities? Or use multiple experimental methods to prove your conclusion.
Response 2: 753b inhibits viability of SCLC cells primarily by inducing apoptotic cell death through degrading BCL-xL and BCL-2, two key anti-apoptotic proteins. The 753b-mediated induction of apoptosis is evident by caspase-3 and PARP cleavage in SCLC cell lines (Fig. 1b-d).
- 753b showed comparable efficacy as DT2216+venetoclax combination to kill BCL-XL/2-dependent SCLC cells. So what are the advantages of 753b compared with DT2216+venetoclax?
Response 3: By degrading both BCL-xL and BCL-2, 753b does not require its combination with venetoclax to effectively target BCL-xL/2 dependent SCLC. This may be advantageous over the combination of DT2216+venetoclax because 753b alone can achieve similar effects at significantly lower doses that can further reduce potential side effects and could be a preferable therapeutic strategy than the combination of DT2216+venetoclax.
- In Figure 1 d, image of BCL-w IB was poor, a more high-quality image should be provided.
Response 4: We have replaced BCL-w IB image in Fig. 1d with a high-quality image.
- The last chart of of Fig4e was mislabeled?
Response 5: We thank the reviewer for noticing it. We have corrected the labelling in the revised manuscript. In addition to Fig.4e, we also corrected y-axis labelling in Fig. 4c and 5d.
- In this paper, the drug 753b was verified by in vivo experiments. The authors should include the detail to determine the drug dosage in experimental method part.
Response 6: We have provided the details of vivo drug dosage in the “Tumor xenograft studies” method.